# Orthogonal intercellular signaling for programmed spatial behavior

Paul K Grant[1,2,†], Neil Dalchau[2,†], James R Brown[1,2], Fernan Federici[1,3], Timothy J Rudge[1], Boyan Yordanov[2], Om Patange[1], Andrew Phillips[2] & Jim Haseloff[1,*]

## Abstract

Bidirectional intercellular signaling is an essential feature of multicellular organisms, and the engineering of complex biological systems will require multiple pathways for intercellular signaling with minimal crosstalk. Natural quorum-sensing systems provide components for cell communication, but their use is often constrained by signal crosstalk. We have established new orthogonal systems for cell–cell communication using acyl homoserine lactone signaling systems. Quantitative measurements in contexts of differing receiver protein expression allowed us to separate different types of crosstalk between 3-oxo-C6- and 3-oxo-C12-homoserine lactones, cognate receiver proteins, and DNA promoters. Mutating promoter sequences minimized interactions with heterologous receiver proteins. We used experimental data to parameterize a computational model for signal crosstalk and to estimate the effect of receiver protein levels on signal crosstalk. We used this model to predict optimal expression levels for receiver proteins, to create an effective two-channel cell communication device. Establishment of a novel spatial assay allowed measurement of interactions between geometrically constrained cell populations via these diffusible signals. We built relay devices capable of long-range signal propagation mediated by cycles of signal induction, communication and response by discrete cell populations. This work demonstrates the ability to systematically reduce crosstalk within intercellular signaling systems and to use these systems to engineer complex spatiotemporal patterning in cell populations.

**Keywords** modeling; quorum sensing; spatial patterning; synthetic biology
**Subject Categories** Signal Transduction; Synthetic Biology & Biotechnology
**Mol Syst Biol. (2016) 12: 849**

## Introduction

The organization of multicellular living systems arises from a hierarchy of interactions. Molecular interactions give rise to changes in gene expression that regulates cell properties and the ability to send and receive intercellular signals. Cell–cell interactions can propagate and undergo feedback and self-ordering, to produce population-level behaviors such as symmetry breaking, cell recruitment, lateral inhibition, and boundary formation. These population-level behaviors emerge from the interplay of processes at different spatial and temporal scales. They underpin the extraordinary levels of self-organization, morphogenesis, and self-repair seen in multicellular organisms. In order to create new types of stable living systems, such as spatially organized microbial populations for bioprocessing or remediation, novel plant structures, and animal tissues or organs, we must be able to engineer these sorts of cellular interactions and harness the emergent properties of self-organization.

Bidirectional intercellular signaling is crucial for creating the interactions that build the feedback mechanisms required for stable patterning. Multicellular patterning mechanisms such as those proposed by Turing (Turing, 1952) and Gierer and Meinhardt (Gierer & Meinhardt, 1972) as well as the creation of tissue organizing centers (Spemann & Mangold, 1924; Struhl & Basler, 1993) all require bidirectional signaling between populations of cells. Acyl homoserine lactone (AHL)-based quorum sensing is one of the simplest known intercellular signaling systems, being comprised of a single biosynthetic enzyme that produces a diffusible small molecule signal, and a single receiver protein that binds the signal and activates transcription (Ng & Bassler, 2009). A large number of different AHL signaling systems are present in nature, which differ in the number of carbons in the acyl side chain and the presence or absence of a ketone group on the third carbon, as well as in the receiver protein that recognizes that AHL (Ng & Bassler, 2009). The use of multiple AHLs in the same system, however, is complicated by the fact that receiver proteins are capable of binding and being activated by AHLs from many different species, and corresponding promoters also share sequence homology (Balagadde *et al*, 2008; Wu *et al*, 2014; Davis *et al*, 2015).

1 Department of Plant Sciences, University of Cambridge, Cambridge, UK
2 Computational Science Laboratory, Microsoft Research, Cambridge, UK
3 Departamento de Genética Molecular y Microbiologia, Facultad de Cs. Biológicas, Universidad Católica de Chile, Santiago, Chile
*Corresponding author. Tel: +44 1223333900; E-mail: jh295@cam.ac.uk
†These authors contributed equally to this work

The two homoserine lactone signaling systems most commonly used in synthetic circuits are the 3-oxo-C6-homoserine lactone (3OC6HSL) receiver, LuxR, from *V. fischeri* (Stevens & Greenberg, 1997) and the 3-oxo-C12-homoserine lactone (3OC12HSL) receiver, LasR from *P. aeruginosa* (Schuster *et al*, 2004). Previous synthetic systems have used both of these receivers simultaneously, but the presence of significant crosstalk between these two systems required either segregating the two receivers in different cells (Brenner *et al*, 2007; Balagadde *et al*, 2008) or incorporating crosstalk into the dynamics of the circuit (Wu *et al*, 2014). Recent work (Chen *et al*, 2015) suggests that C4- and 3O-C14-HSL can be used orthogonally but it is unclear the extent to which crosstalk does or does not exist between these signals. In order to engineer two-channel receiver devices with minimal crosstalk, we first quantified this crosstalk by expressing receiver proteins in varying combinations and levels. We then made base pair changes in the pLux promoter to minimize noncognate receiver protein binding while maximizing cognate receiver protein binding. This quantification allowed us to infer parameters for a detailed mathematical model of the system, which allowed us to make predictions about the optimal expression level of each receiver protein and to evaluate our devices against these predictions. In doing so, we have systematically reduced crosstalk to produce a device that differentiates between two different AHL inputs in the same cell and produces two orthogonal outputs. We used this device in a novel spatial assay system that allowed us to place populations in arbitrary geometries and precisely measure gene expression. This enabled us to engineer bidirectional population-level feedback interactions that resulted in autoinduction and long-range propagation of a signal.

# Results

## Quantitative measurement of crosstalk

In order to quantify the crosstalk between 3OC6HSL and 3OC12HSL signaling systems, we built a set of receiver devices (Fig 1A) that allowed the activity of the wild-type Lux promoter pLux (Bba_R0062) to be measured as a ratio of the fluorescence output from pLux to that of a reference promoter (Brown, 2013; Yordanov *et al*, 2014; Rudge *et al*, 2015). The devices constitutively expressed either LuxR, LasR, or both, in a bicistronic operon driven either by a strong synthetic promoter pLlacO1 (Bba_R0011) or a weak promoter pCat (Bba_I14033). The fluorescent protein used for measurement was eYFP, expressed under the control of pLux. The reference fluorescent protein used was eCFP, expressed constitutively by the bacteriophage lambda promoter pR (Bba_R0051; Fig 1A). By measuring the fluorescence output over time from these devices in a microplate fluorometer assay, we were able to quantify the activity of the pLux promoter in a highly accurate and reproducible manner. Ratiometric promoter activity, as described in Brown (2013) and Rudge *et al* (2015), provides a measurement that is robust to varying growth conditions, since the reference channel provides a measure of extrinsic variation. Without such a control, our ability to reproducibly measure the intrinsic characteristics of promoters would be greatly reduced, diminishing the accuracy of a quantitative model of crosstalk. In previous work, we used Bba_J23101 as our reference promoter (Brown, 2013; Yordanov *et al*, 2014; Rudge

*et al*, 2015). However, Bba_J23101 is inhibited in the presence of 3OC12HSL-LasR (Appendix Fig S1), requiring us to use a new reference pR (Bba_R0051). This alternative reference promoter is not inhibited by 3OC12HSL-LasR (Appendix Fig S1A) and provides comparable measurements to those obtained with Bba_J23101 (Appendix Fig S1B).

We measured the response of the pLux promoter to varying concentrations of its cognate signal 3OC6HSL, in the device strongly expressing both receiver proteins using the constitutive promoter pLlacO1 (plasmid pR0011LL123, Appendix Table S1). We observed a maximal activity of about 10 relative promoter units (RPU), meaning that the activity was 10 times that of the reference promoter, and half-maximal activity, at about 5 nM (Fig 1B, blue points). The response to the interfering signal 3OC12HSL showed a similar maximal activity, but ~100-fold lower sensitivity (Fig 1B, red points). This magnitude of crosstalk is similar to what has been previously measured and would not allow for orthogonal signaling (Wu *et al*, 2014). However, the device expressing both receiver proteins at a low level using the constitutive promoter pCat (pCatLL123) displayed a maximal response to 3OC12HSL that was ~10-fold lower (Fig 1C, red points) while maintaining the maximal response to 3OC6HSL, although the sensitivity was ~100-fold lower (Fig 1C, blue points). This suggested that simply by manipulating the expression level of the receiver proteins, a large influence on crosstalk could be achieved.

## A mathematical model of signal crosstalk

In order to use the experimental data to optimize the system design, we built a quantitative model of signal crosstalk and inferred its parameters such that the model was able to fit all of the available ratiometric receiver data. This allowed us to understand how changes in components affected the entire system. The model simulates an equilibrium response to concentrations of 3OC6HSL and 3OC12HSL, which depends both on the intracellular abundance of LuxR and LasR, and the affinity relationships between the promoter, the receiver proteins and the HSL signals. A detailed derivation is provided in Appendices B and C. Briefly, we started from a system of chemical reactions that describes binding/unbinding, transcription, translation, degradation and growth dilution.

A LuxR module involves the reactions

$$\varnothing \to R, R + C_k \leftrightarrow R_k, R_k + R_k \leftrightarrow D_k, G + D_k \leftrightarrow G.D_k,$$
$$G.D_k \to G.D_k + \text{mRNA}$$

where $R$ represents LuxR, $G$ represents the promoter of a gene, $C$ represents HSL, and the subscript $k$ denotes either 3OC6HSL (6) or 3OC12HSL (12). Correspondingly reactions for LasR ($S$) are given by

$$\varnothing \to S, S + C_k \leftrightarrow S_k, S_k + S_k \leftrightarrow E_k, G + E_k \leftrightarrow G.E_k,$$
$$G.E_k \to G.E_k + \text{mRNA}$$

Growth dilution at rate $\gamma$ is modeled by

$$R, S, R_k, S_k, D_k, E_k, G, G.D_k, G.E_k \xrightarrow{\gamma} \varnothing$$

We assume a zero-order production rate for gene $G$, which models the replacement of plasmids during cell division. This is

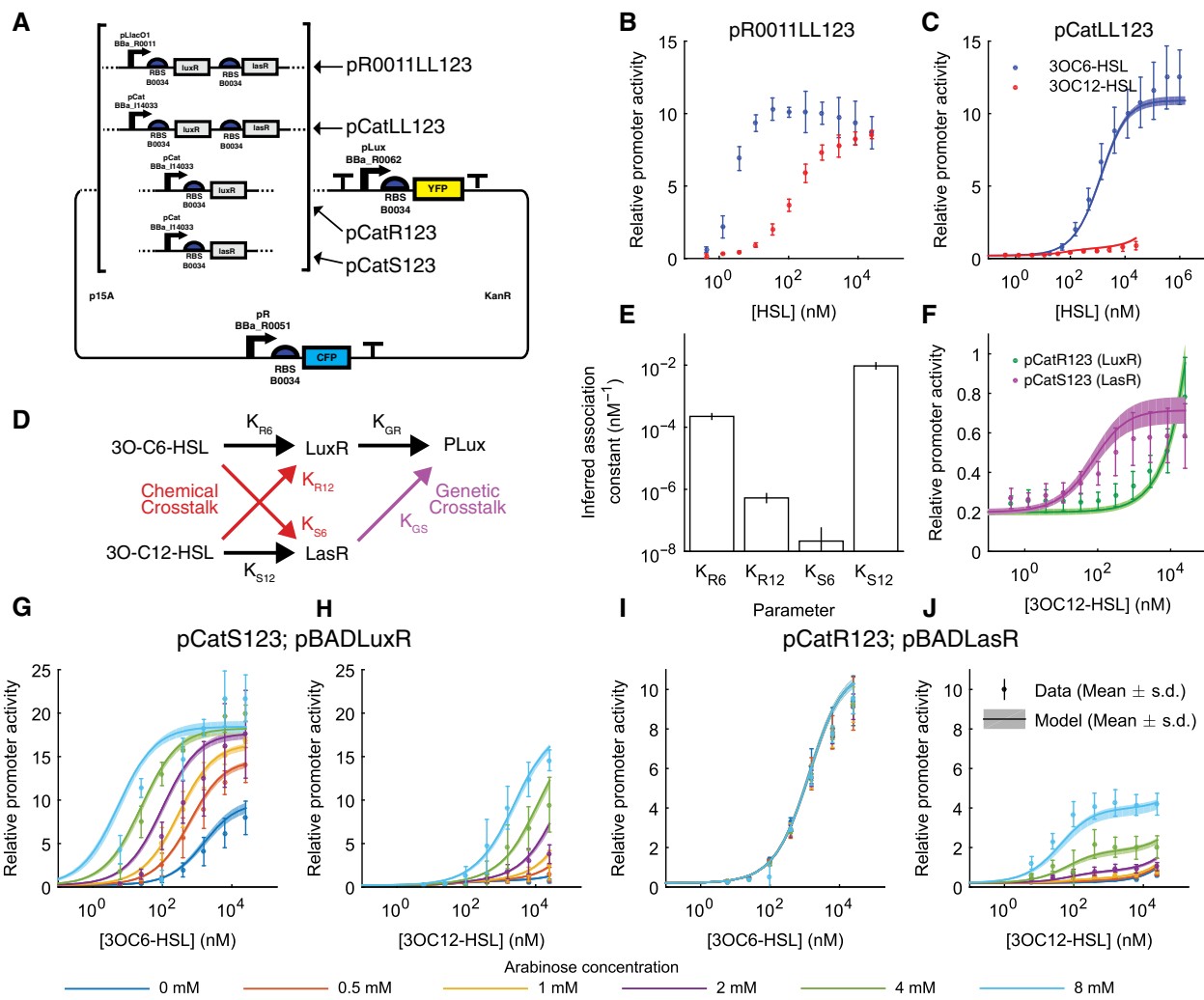

**Figure 1.  The wild-type lux promoter exhibits both chemical and genetic crosstalk, which is strongly dependent on receiver protein expression level.**

A    Ratiometric reporter constructs express eYFP under the control of pLux (Bba_R0062) and eCFP under pR (Bba_R0051) but differ in which receiver proteins are expressed and the strength of their expression (described in Appendix Table S1). Strong expression is driven by pLlacO1 (Bba_R0011) and weak expression by pCat (Bba_I14033).

B    The relative activity of pLux in the presence of strong expression of both receiver proteins (plasmid pR0011LL123) as a function of 3OC6HSL concentration (blue points) and 3OC12HSL concentration (red points).

C    The relative activity of pLux in the presence of weak expression of both receiver proteins (plasmid pCatLL123) as a function of 3OC6HSL concentration (blue points) and 3OC12HSL concentration (red points).

D    Crosstalk can occur due to receiver protein binding noncognate HSL (chemical crosstalk) or due to signal-bound receiver activating transcription at a noncognate promoter (genetic crosstalk).

E    Inferred association constants for LuxR or LasR with 3OC6HSL or 3OC12HSL

F    The relative activity of pLux in the presence of weak expression of LuxR (plasmid pCatR123, green points and line) or weak expression of LasR (plasmid pCatS123, magenta points and line) as a function of 3OC12HSL concentration.

G–J    The relative activity of pLux in the presence of weak expression of LasR and inducible expression of LuxR (G, H, plasmids pCatS123 and pBADLuxR) or weak expression of LuxR and inducible expression of LasR (I, J, plasmids pCatR123 and pBADLasR) as a function of 3OC6HSL concentration (G, I) or 3OC12HSL concentration (H, J). Inducible expression was varied via arabinose concentration as indicated by the color code.

Data information: (B, C, F–I) Relative promoter activity (ρ = deYFP/deCFP) with respect to the reference pR is reported as a function of HSL concentration. Points indicate the mean of three replicates and error bars indicate the standard deviation while lines and shading indicate the mean and standard deviation of the best-fit models, respectively.

Source data are available online for this figure.

motivated by wanting to balance plasmid replication with dilution in equilibrium. Finally, we note that transcription and translation of LuxR/LasR are lumped into a single generation reaction. This is for simplicity, as we will seek an equilibrium eventually, and the LuxR/

LasR factors in the model will become subsumed into a single parameter.

From the reactions, we derived a corresponding system of ordinary differential equations (ODEs), then solved for the equilibrium

solutions, simplifying the parameterization into affinity relationships (Appendix B). For this *Full model*, it was not possible to derive a closed-form expression (a single equation purely as a function of the parameters) for the transcription rate. Therefore, to evaluate the equilibrium response of the Full model, a numerical approach was required. However, by assuming that the effect of dilution was negligible for some molecular complexes, we were able to derive an approximate closed-form expression for the transcription rate, as a function of the 3OC6HSL and 3OC12HSL concentration:

$$f(C6,C12) = \frac{a_0 + a_1^R K_{GR} r^2 \frac{K_{R6}^n C_6^n + K_{R12}^n C_{12}^n}{(1+K_{R6}C_6+K_{R12}C_{12})^n} + a_1^S K_{GS} s^2 \frac{K_{S6}^n C_6^n + K_{S12}^n C_{12}^n}{(1+K_{S6}C_6+K_{S12}C_{12})^n}}{1 + K_{GR} r^2 \frac{K_{R6}^n C_6^n + K_{R12}^n C_{12}^n}{(1+K_{R6}C_6+K_{R12}C_{12})^n} + K_{GS} s^2 \frac{K_{S6}^n C_6^n + K_{S12}^n C_{12}^n}{(1+K_{S6}C_6+K_{S12}C_{12})^n}}$$

(1)

This approximate "Simplified model" has the advantages of (i) allowing a more direct interpretation of the functional response to different levels of HSL and receiver protein, and (ii) being computationally more efficient to evaluate.

Appendix C contains the derivation of the Simplified model and Appendix Table S6 provides a complete description of the parameters. To summarize the parameter definitions, $K_{Ri}$ is the binding affinity of LuxR to signal $i$, $K_{Si}$ is the equivalent parameter for LasR, $K_{GR}$ and $K_{GS}$ describe the binding of LuxR/LasR-based regulator complexes to the pLux promoter, $a_1$ is the maximal transcription rate from a promoter bound to a LuxR/LasR-based regulator complex, and $a_0$ is the basal transcription rate. The quantities $r$ and $s$ indicate the intracellular LuxR and LasR levels, relative to constitutive expression with pCat.

As we found only minor differences in the behavior of the Full and Simplified models, all subsequent analysis was conducted using the Simplified version, and results from the Simplified model alone are presented in the main text of the manuscript. Equivalent behaviors of the Full model can be found in the Appendix Figs S2–S5.

### Identifying the sources of crosstalk

The measurable response of the wild-type promoter pLux to 3OC12HSL in the context of weak expression of receiver protein meant that reducing receiver protein expression diminished but did not eliminate crosstalk. The remaining nonspecific response to 3OC12HSL could come from 3OC12HSL binding to its noncognate receiver LuxR, and activating transcription through the cognate promoter for LuxR, which we refer to as "chemical crosstalk". Alternatively, the unintended response could come from 3OC12HSL binding to its cognate receiver LasR, and activating transcription through a noncognate promoter for LasR, which we refer to as "genetic crosstalk" (Fig 1D). To determine the source of the crosstalk, we measured the pLux response to 3OC12HSL in devices expressing either LuxR (pCatR123) or LasR (pCatS123) alone. Both devices responded to 3OC12HSL with a maximal activity of ~0.7 RPU, indicating that both chemical and genetic crosstalk were present (Fig 1F). Interestingly, while the LuxR-expressing device responded to 3OC6HSL with the expected maximal expression of 10 RPU, the LasR-expressing device did not respond to 3OC6HSL at any concentration tested (Appendix Fig S4) suggesting that this source of chemical crosstalk is not present. The relative affinity of each of the

receiver proteins for each of the HSLs can be expressed in the form of an inferred association constant (Fig 1E). The affinity of LuxR for 3OC6HSL is ~$10^2$ times greater than its affinity for 3OC12HSL, while the affinity of LasR for 3OC12HSL is ~$10^6$ times greater than its affinity for 3OC6HSL, again suggesting that the LuxR-3OC12HSL chemical crosstalk is significantly greater than the chemical crosstalk for LasR-3OC6HSL.

### Quantifying the influence of receiver protein expression level

To determine the degree to which chemical crosstalk could be managed by changing receiver protein expression level, and to gain a more precise description of the functioning of the system, we expressed each receiver protein under the control of an inducible promoter (AraC/pBAD, BBa_I0500; Guzman *et al*, 1995). Separate high copy plasmids were used to inducibly express one receiver protein, in combination with receiver devices constitutively expressing the other receiver protein. We used concentrations of arabinose from 0 to 8 mM to vary the expression of one receiver protein while keeping the other constant. In each of these conditions, we measured the response of the system to concentrations of 3OC6HSL and 3OC12HSL ranging from 10 nM to 25 μM. Using this data to parameterize the model allowed us to infer a functional relationship between arabinose concentration and a relative concentration of each receiver protein (see Appendix B.4). Increasing expression of LuxR resulted in a response to 3OC6HSL that included a ~twofold increase in maximal transcription level (within the range of 3OC6HSL measured) and a ~100-fold increase in sensitivity (Fig 1G). Increasing LuxR expression also resulted in an increased response to 3OC12HSL including a ~20-fold increase in maximal transcription (Fig 1H). Increasing LasR expression, however, had no effect on the response to 3OC6HSL (Fig 1I) while increasing the response to 3OC12HSL by increasing the maximal transcription ~sixfold without altering the sensitivity (Fig 1J). This is consistent with a very low affinity of LasR for 3OC6HSL and thus no chemical crosstalk in this direction. This indicated that, by expressing LuxR at the lowest level that still maintained an appropriate response, we would be able to significantly reduce the effects of chemical crosstalk while still being free to express LasR at a high level to allow for maximal sensitivity.

### Mutant orthogonal promoters reduce genetic crosstalk

Achieving orthogonal responses to both 3OC6HSL and 3OC12HSL within the same cell requires two distinct promoters that respond independently to each signal. To identify suitable promoters with minimal genetic crosstalk, we examined the pLux promoter to identify targets for mutations that might differentiate between responses to LuxR and LasR. The consensus-binding sequence for LasR recognition is not fully consistent between studies but a CT dinucleotide at base pairs 3–4 and an AG at base pairs 17–18 are the most commonly cited as required for binding (Whiteley & Greenberg, 2001; Schuster *et al*, 2004; Gonzalez-Valdez *et al*, 2014). These four nucleotides are shared with the consensus LuxR binding sequence (Fig 2A) (Antunes *et al*, 2008). We made 7 putative LuxR-specific promoters by making single base pair changes at positions 3, 4, 17, and 18 in order to disrupt LasR binding (Fig EV1). We made 5 putative LasR-specific promoters by making single and double base pair

changes at positions 5 and 16 in order to disrupt LuxR binding without disrupting LasR binding (Fig EV1A). We made ratiometric receiver devices with all of these promoters driving eYFP and with constitutive LuxR or LasR driven by promoter pLlacO1. We measured the response to 3OC6HSL in the LuxR-containing devices and the response to 3OC12HSL in the LasR-containing devices (Fig EV1B and C). We chose the two promoters (pLux76 and

pLas81) that minimized the response to the heterologous signal and used them for further device construction (Figs 2A and EV1).

We built ratiometric receiver devices containing both receiver proteins expressed bicistronically under the pCat promoter, along with either pLux76 or pLas81 driving eYFP (pCatLL76 and pCatLL81, respectively). pLux76 displayed a ~fivefold lower maximal transcription in response to 3OC6HSL compared to the

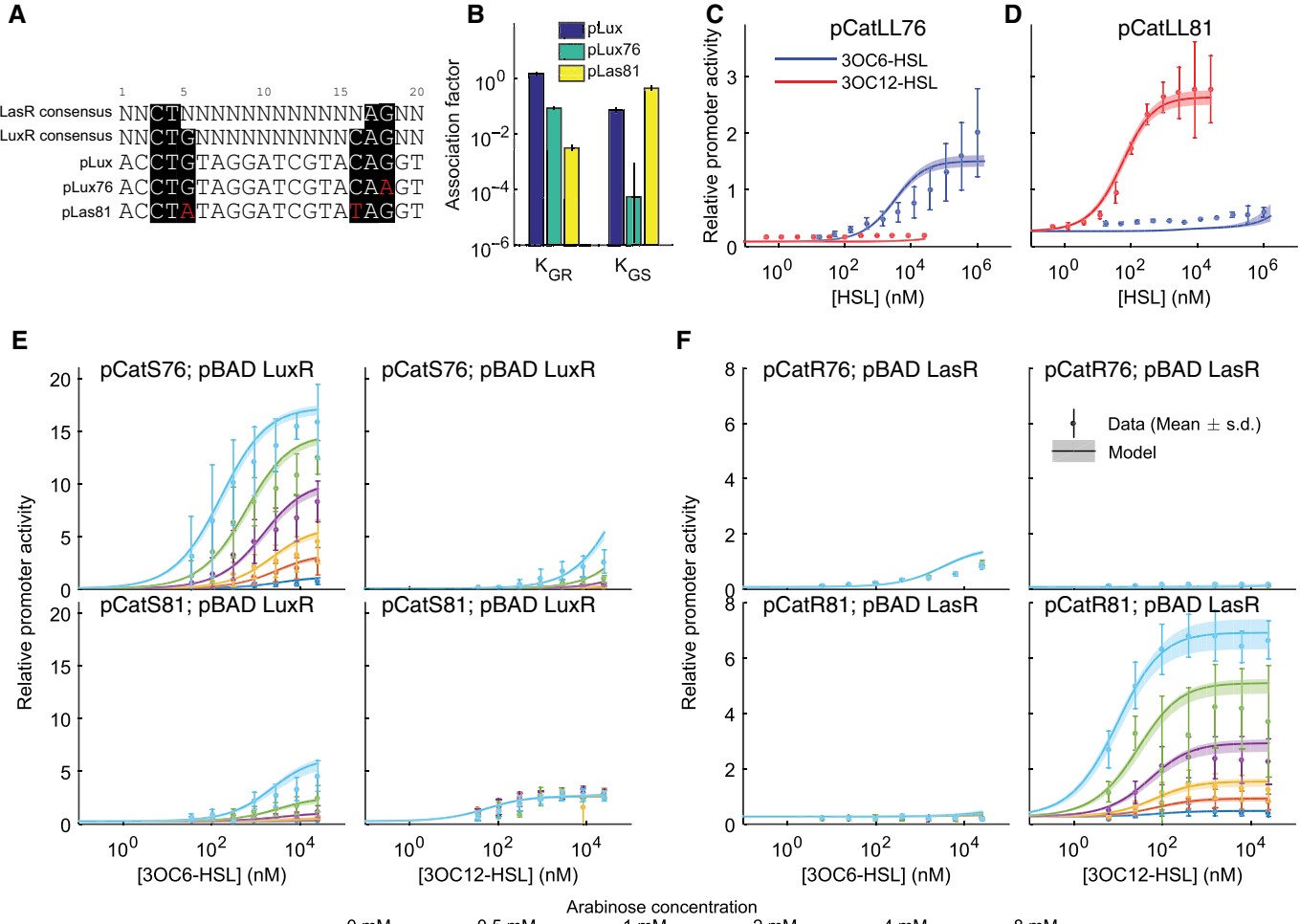

**Figure 2. Base pair changes in the Lux promoter can produce 3OC6HSL- or 3OC12HSL-specific responses.**

A   A sequence comparison between the Lux box and a consensus LasR recognition sequence suggests targets for base pair changes to generate specificity. Consensus-binding sequences are highlighted in black while specific mutations are in red. pLux76 and pLas81 were chosen for greatest minimization of crosstalk from a group of potential specific promoters.

B   Inferred association constants of LuxR ($K_{GR}$) or LasR ($K_{GS}$) with pLux, pLux76, or pLas81.

C   The relative activity of pLux76 in the presence of weak expression of both receiver proteins (plasmid pCatLL76) as a function of 3OC6HSL (blue points) or 3OC12HSL (red points).

D   The relative activity of pLas81 in the presence of weak expression of both receiver proteins (plasmid pCatLL81) as a function of 3OC6HSL (blue points) or 3OC12HSL (red points).

E   The relative activity of pLux76 (top) or pLas81 (bottom) in the presence of weak expression of LasR and inducible expression of LuxR (plasmids pCatS76 and pBADLuxR or plasmids pCatS81 and pBADLuxR) as a function of 3OC6HSL concentration (left) or 3OC12HSL concentration (right). Inducible expression was varied via arabinose concentration as indicated by the color code.

F   The relative activity of pLux76 (top) or pLas81 (bottom) in the presence of weak expression of LuxR and inducible expression of LasR (plasmids pCatR76 and pBADLasR or plasmids pCatR81 and pBADLasR) as a function of 3OC6HSL concentration (left) or 3OC12HSL concentration (right). Inducible expression was varied via arabinose concentration as indicated by the color code.

Data information: In (C–F), points indicate the mean of three replicates and error bars indicate the standard deviation while lines and shading indicate the mean and standard deviation of the best-fit models, respectively.

Source data are available online for this figure.

wild-type pLux with the same receiver expression (Fig 1D), but the response to 3OC12HSL was now low enough to be undetectable (Fig 2C). Furthermore, pLas81 displayed a ~fivefold increase in maximal transcription in response to 3OC12HSL while also diminishing the response to 3OC6HSL to undetectably low levels (Fig 2C). In the mathematical model, affinities of each of the receiver proteins for each of the promoters were expressed in the form of inferred association constants $K_{GR}$ and $K_{GS}$ for the wild-type pLux and each of the mutant promoters (Fig 2B). The mutations to pLux76 reduced its affinity to LuxR ($K_{GR}$) by more than tenfold compared to the wild type, but also reduced its affinity to LasR ($K_{GS}$) to levels indistinguishable from 0 (Fig 2B). In contrast, pLas81 displayed an almost tenfold greater affinity for LasR compared to pLux, while its affinity for LuxR was reduced more than ~100-fold, resulting in a signal to crosstalk ratio of ~$10^2$.

## Tuning receiver protein expression

The mutant promoters successfully minimized genetic crosstalk, but also altered the signal response. As a result, some tuning of the receiver protein expression was required to minimize chemical crosstalk while maximizing signal response. To determine the optimal receiver protein expression levels, we again used arabinose-inducible receiver protein expression. Increased LuxR expression via increased arabinose concentration in the pLux76-containing device resulted in up to a ~sevenfold increase in maximal transcription in response to 3OC6HSL (Fig 2E, top left). As expected, we also saw an increase in chemical crosstalk, in the form of a ~fourfold increase in maximal transcription in response to 3OC12HSL at 8 mM arabinose (Fig 2E, top right, light blue points). LuxR induction with 8 mM arabinose also resulted in increased genetic crosstalk, in the form of an ~eightfold increase in maximal transcription through pLas81 in response to 3OC6HSL (Fig 2E, bottom left, light blue points). Interestingly, changing the expression of LuxR had no effect on the response of pLas81 to 3OC12HSL (Fig 2E, bottom right) indicating that the decreased affinity of pLas81 for LuxR was sufficient to render a combined genetic and chemical crosstalk response undetectable at these expression levels. Together, these data suggest

that there is an optimal level of LuxR expression at which the desired response (by pLux76 to 3OC6HSL) is maximized, while both the genetic and chemical crosstalk are minimized.

Increased expression of LasR resulted in a ~10-fold increase in maximal expression by pLas81 in response to 3OC12HSL (Fig 2F, bottom right). In contrast to LuxR, however, increased expression of LasR did not result in increased chemical or genetic crosstalk in the form of expression by pLas81 in response to 3OC6HSL, or by pLux76 in response to either HSL (Fig 2F). These results demonstrate that we have successfully decreased the affinity of pLux76 for 3OC12HSL-LasR compared to the wild-type Lux promoter, and also suggest that the affinity of LasR for 3OC6HSL is very low.

## Construction of an optimal two-channel receiver device

With these components characterized, we were then able to construct two-channel receivers that would respond to the two AHL signals with orthogonal outputs. We built these receivers on the same backbone as our ratiometric receivers, expressing eYFP under pLas81 and eCFP under pLux76 (Fig 3A). In order to characterize these devices, we created a strain containing a chromosomally integrated mRFP1 that we used as a reference for ratiometric measurements. The mRFP1 measurement provided a reference channel for ratiometric liquid culture experiments and also served as a proxy for cell density in later solid culture experiments. An optimal two-channel receiver would maximize the signal through each intended channel while minimizing crosstalk. Because we had a fully parameterized model (Eqn 1) that was consistent with all liquid culture data measuring pLux, pLux76, and pLas81 across a range of LuxR/LasR expression levels (Figs 1 and 2), we were able to predict optimal LuxR/LasR expression levels that minimize crosstalk. We defined the optimal response as a maximization of the signal to crosstalk ratio

$$\frac{\text{CFP}_{C6} \times \text{YFP}_{C12}}{\text{CFP}_{C12} \times \text{YFP}_{C6}} \tag{2}$$

where $\text{XFP}_{CY}$ is the ratiometric expression level of fluorescent protein X at 100 nM of HSL Y. By plotting this quantity as a function

Figure 3.    **Expressing receiver proteins at optimal levels minimizes crosstalk while maintaining sensitivity in dual-channel reporter constructs.**

A   Double reporter constructs express eYFP under the control of pLas81 and eCFP under pLux76. Receiver proteins are expressed under the control of pCat (Bba_I14033), pLlacO1 (Bba_R0011), or pLTetO1 (Bba_R0040) and also vary in the RBS used to control translation.

B   A fully parameterized model is used to predict the optimal expression levels for LuxR and LasR. The point at which expression levels of LuxR and LasR result in the maximal simulated signal to crosstalk ratio is labeled with a dot while isolines are colored to represent lower values of that ratio. Double reporters expressing eCFP under the control of pLux76 and eYFP under the control of pLas81 along with both receiver proteins under the control of various promoters and RBS sequences (see Appendix Table S2) are each represented with an "X" placed at the expression levels of LuxR and LasR that were inferred from measuring the plasmid's response to 3OC6HSL and 3OC12HSL in both the eCFP and eYFP channels (ratiometrically with respect to signal from a chromosomally integrated constitutively expressed mRFP1). The color of the X reflects the actual ratio of signal to crosstalk of the data for that receiver, on the same scale as the isolines. Expression levels of LuxR and LasR are normalized to the levels of expression in the ratiometric reporters driven by pCat.

C   Constitutive receiver protein concentration in double reporters is equivalent to that of inducible expression at interpolated arabinose concentrations. Shown are the best-fit relationships between LuxR (green) or LasR (pink) concentration and arabinose for all experiments involving pBAD-LuxR and pBAD-LasR (lines), with standard deviations computed from 5,000 MCMC samples (shading). Also indicated are the relative LuxR/LasR concentrations inferred for double reporter constructs in pCat units.

D   Activity (relative to chromosomal constitutive mRFP1) of pLux76 (eCFP, blue) and pLas81 (eYFP, red) in the pR33S175 construct as a function of 3OC6HSL (top) or 3OC12HSL (bottom). Points indicate the mean of three replicates and error bars indicate the standard deviation while lines indicate the mean of the best-fit models. Simulations used LuxR and LasR levels indicated in (A) ($r$ = 5.89, $s$ = 2.97), and all other parameters as specified in Appendix Table S6.

E   Image at $t$ = 1,500 min of chromosomal constitutive mRFP1 cells containing each of the double reporters or a control construct constitutively expressing eCFP and eYFP (pPRYFPPRCFP), plated on a membrane printed with a hydrophobic grid along with 3OC6HSL sender cells and 3OC12HSL sender cells.

F   Activity (relative to chromosomal constitutive mRFP1) of pLux76 (eCFP, top) and pLas81 (eYFP, bottom) for each double reporter is plotted against time for every other grid square according to the color scheme shown. Experimental data are plotted as a solid line while model simulation is plotted as a dotted line.

Source data are available online for this figure.

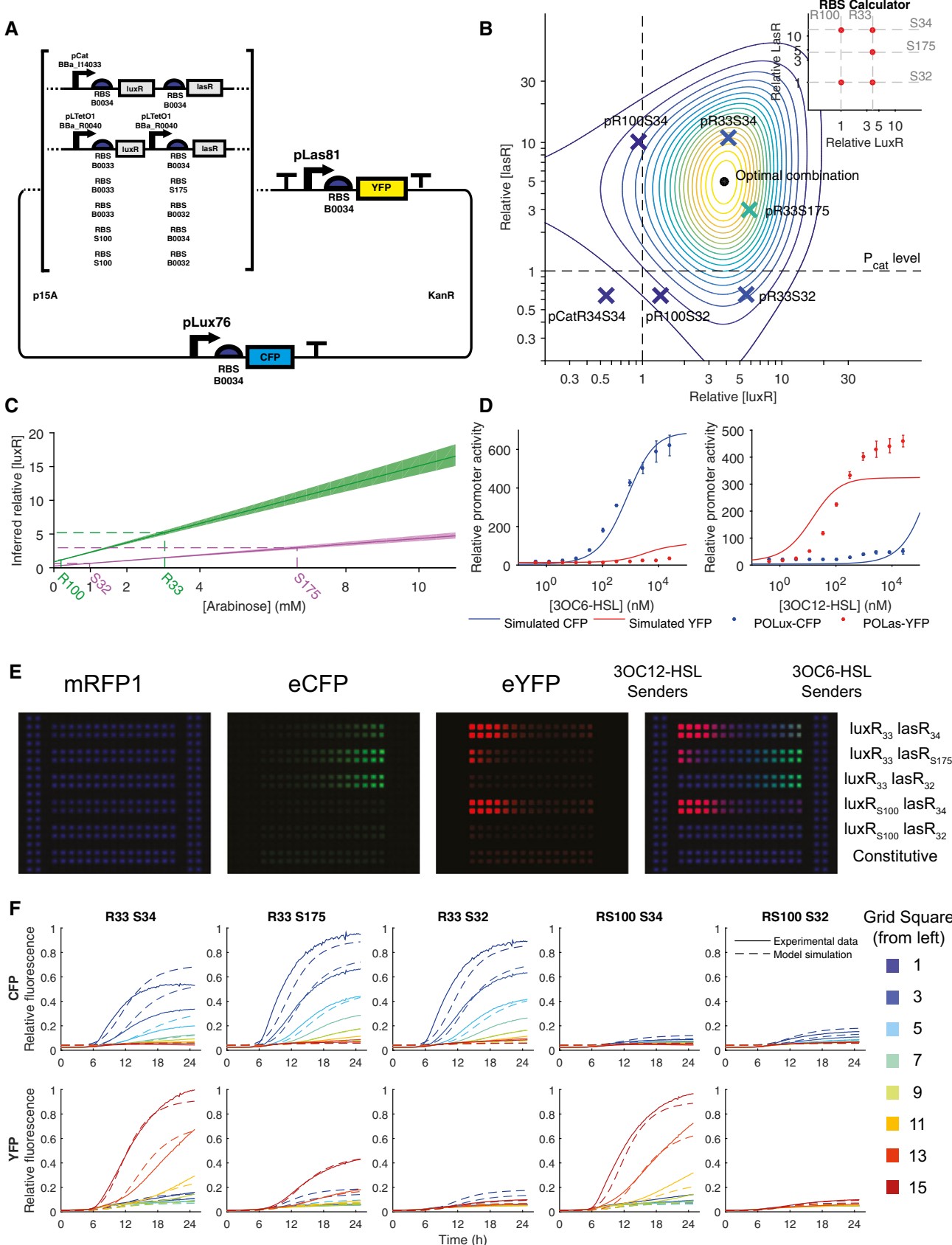

**Figure 3.**

of LuxR and LasR concentrations, we determined the concentrations that would give us the maximal signal/crosstalk ratio and therefore a target for the optimal expression level of each receiver protein (Fig 3B; see Appendix E for further details).

Because it was not straightforward to design a construct that would constitutively produce the desired optimal receiver protein expression levels, we made a series of devices that varied the ribosome-binding site sequence controlling the translation of both LuxR and LasR. We used the RBS calculator described in (Salis *et al*, 2009) to predict the relative translation rates of LuxR and LasR (Fig 3B, inset). For each device, we measured eCFP and eYFP output in response to 3OC6HSL and 3OC12HSL (as a ratio with respect to the mRFP1 signal, Appendix Fig S8). We then used these data to estimate the intracellular LuxR and LasR concentrations in each device (Fig 3B and C). The inferred levels of LuxR and LasR in our first four designs (pR100S32, pR100S34, pR33S32, and pR33S34) matched the relative predictions of the RBS calculator (Fig 3B and inset, quantified in Appendix Table S7) but comparison to the optimization plot suggested that an intermediate expression of LasR along with the LuxR expression from RBS R33 would maximize the signal to crosstalk ratio. We used the RBS calculator to design an RBS (S175) that would achieve this expression level. For this construct (pR33S175) we used a variant of LasR that contains an E11K substitution. This variant has similar properties to the wild-type LasR (Appendix Fig S9). Inferred expression levels in pR33S175 matched predictions (Fig 3B and inset) and the fluorescence responses to each signal showed very little crosstalk (Fig 3D) so we chose to use it for further experiments. In response to 3OC6HSL, our optimized double receiver (pR33S175) displayed a maximal activity of 600 RPU (relative to chromosomal mRFP1), half-maximal activity at 25 nM in the eCFP channel, and no measurable increase in activity in the eYFP channel (Fig 3C). In response to 3OC12HSL, it displayed a maximal activity of 500 RPU, half-maximal activity at 10 nM in the eYFP channel, and a maximal activity of 30 RPU in the eCFP channel (Fig 3C). By combining modeling, data collection, and rational design we were able to arrive at a near-optimal double receiver device with minimal iterations of the design, build, and test loop (Andrianantoandro *et al*, 2006).

### A novel system for arranging and measuring spatially discrete cell populations on solid media

In addition to a two-channel signaling system, engineering cellular interactions required control over the geometry of populations so that spatial parameters such as diffusion coefficients could be inferred. To this end we developed a novel spatial assay system using commercially available membranes printed with hydrophobic grids. Membranes were placed on solid media, inoculated with dilute cultures, and incubated in a macroscopic fluorescence imaging system of our own design (Fig EV2D). This system allowed us to maintain genetically distinct populations in arbitrary regular geometries, keeping each population constrained within the square in which it was inoculated and to image fluorescent output over time. We could then treat each grid square as an independent population (similar to the wells in previous microplate experiments), but with the important property that while cells themselves were maintained in separate populations, signals were free to diffuse between neighboring populations. This allowed us to observe

changes in fluorescence over time in response to the changes in signal distribution due to production and diffusion. By using our chromosomal mRFP1-expressing strain, we could normalize output signal to the gene-expression capacity of a population within a square of the grid (primarily determined by the number of cells present) in order to compare results between liquid culture microplate experiments and solid media imaging experiments.

To measure the response of our two-channel receiver devices to physiologically relevant concentrations of signals, we plated cells containing each device in 2 rows of grid squares and then plated cells containing sender devices in columns adjacent to the rows (Fig 3D). The 3OC6HSL sender device consisted of the 3OC6HSL synthase (LuxI), driven by a constitutive promoter (Bba_R0011). The 3OC12HSL sender device consisted of the 3OC12HSL synthase (LasI), driven by pBAD under constant arabinose induction (25 mM arabinose). All cells were of the chromosomal mRFP1-expressing strain. Images taken 1,500 min after plating (Fig 3D) showed that the two-channel receiver devices displayed the same relative strengths of signal and crosstalk as seen in the liquid culture experiments. The optimized double receiver (pR33S175) displayed a strong response in the appropriate signal channel with undetectable crosstalk (Fig 3D). Traces of average normalized fluorescence in each grid square plotted against time (Fig 3E, solid lines) allowed us to track the evolution of response of each population as the signals diffused. The mRFP1 traces could be modeled with a Gompertz growth model with a good fit (Appendix Fig S10), suggesting that the mRFP1 was a good proxy for growth. By using the growth model parameterized so as to fit the mRFP1 data, along with the parameterized model inferred from liquid culture experiments (Eqn 1; Appendix Table S6), we were able to model the response of the receiver devices in time and space (see Appendix F.1). Using this model we inferred parameters for the production and diffusion of AHLs (Appendix Table S8) which allowed us to produce simulations of the behavior of each of the devices in each of the grid squares over time (Fig 3E, dotted lines).

### Coupling input signals to orthogonal outputs with signal relay devices

The final components that were required for engineering bidirectional cellular interactions were devices capable of both sending and receiving signals while maintaining their orthogonality. In order to achieve this, we built relay devices that send one AHL in response to receiving the other. Similar sending devices have been constructed previously (Brenner *et al*, 2007) using C4- and 3OC12HSL. However, since crosstalk was still present in those circuits, it was necessary for each cell to contain only one receiver protein, preventing detection in both channels. In contrast, we were able to send and receive simultaneously on two channels in the same cell. Our devices consist of the synthases LuxI or LasI driven by pLas81 or pLux76, respectively, using ribosome-binding sites that were selected by iterative screening as described previously (Fig 3).

We transformed cells containing the optimized double receiver (pR33S175) with each of these devices, and plated them on membranes printed with hydrophobic grids in wells of 8-well plates containing varying concentrations of 3OC6HSL and 3OC12HSL. These cells occupied the first two rows of grid squares in each well (Figs 4 and EV2). Adjacent to these we plated receivers alone (transformed with an empty second plasmid for antibiotic resistance).

Cells containing the double receiver and the pLas81-LuxI relay device, as well as the adjacent double receivers with empty vector, were induced by 16–1,600 nM 3OC12HSL. The fluorescence response in the YFP channel was consistent with direct activation by the 3OC12HSL in the media in a concentration-dependent manner (Fig 4A). Plotting accumulation of eYFP fluorescence over time revealed that the populations of all of the grid squares in each condition behaved equivalently.

In order to increase the accuracy of the model in this context, it was necessary to have a more accurate representation of the response of pLas81 to 3OC12HSL in pR33S175 (Fig 3D). We used a Hill function to fit the liquid culture data (Appendix F.2; Appendix Fig S11), and used this transfer function in all subsequent modeling. A model of the relay devices (Appendix F.3), with growth calibrated to the mRFP1 fluorescence (Appendix Fig S12), very

closely matched the measured fluorescence when the synthesis rates of LuxI, LasI, 3OC6HSL and 3OC12HSL were selected appropriately (Fig 4C, Appendix Fig S13, Appendix Table S9). Fluorescence in the CFP channel displayed a gradient of intensity, with the highest response observed in the cells containing the relay device and diminishing with distance from those cells. This suggested that the relay was indeed functioning as a sender in the appropriate channel (Fig 4A and C). Induction with 1,000 nM 3OC6HSL resulted in uniform activation only in the CFP channel, indicating that there was no detectable crosstalk in sending or receiving (Fig 4A). The pLux76-LasI device in the same assay, induced with 16–1,600 nM 3OC6HSL, showed the same response but in the opposite fluorescence channels—constant response in the CFP channel, and a graded sending–receiving response in the YFP channel (Fig 4B and D). The noncognate inducer (3OC12HSL) induced uniform fluorescence

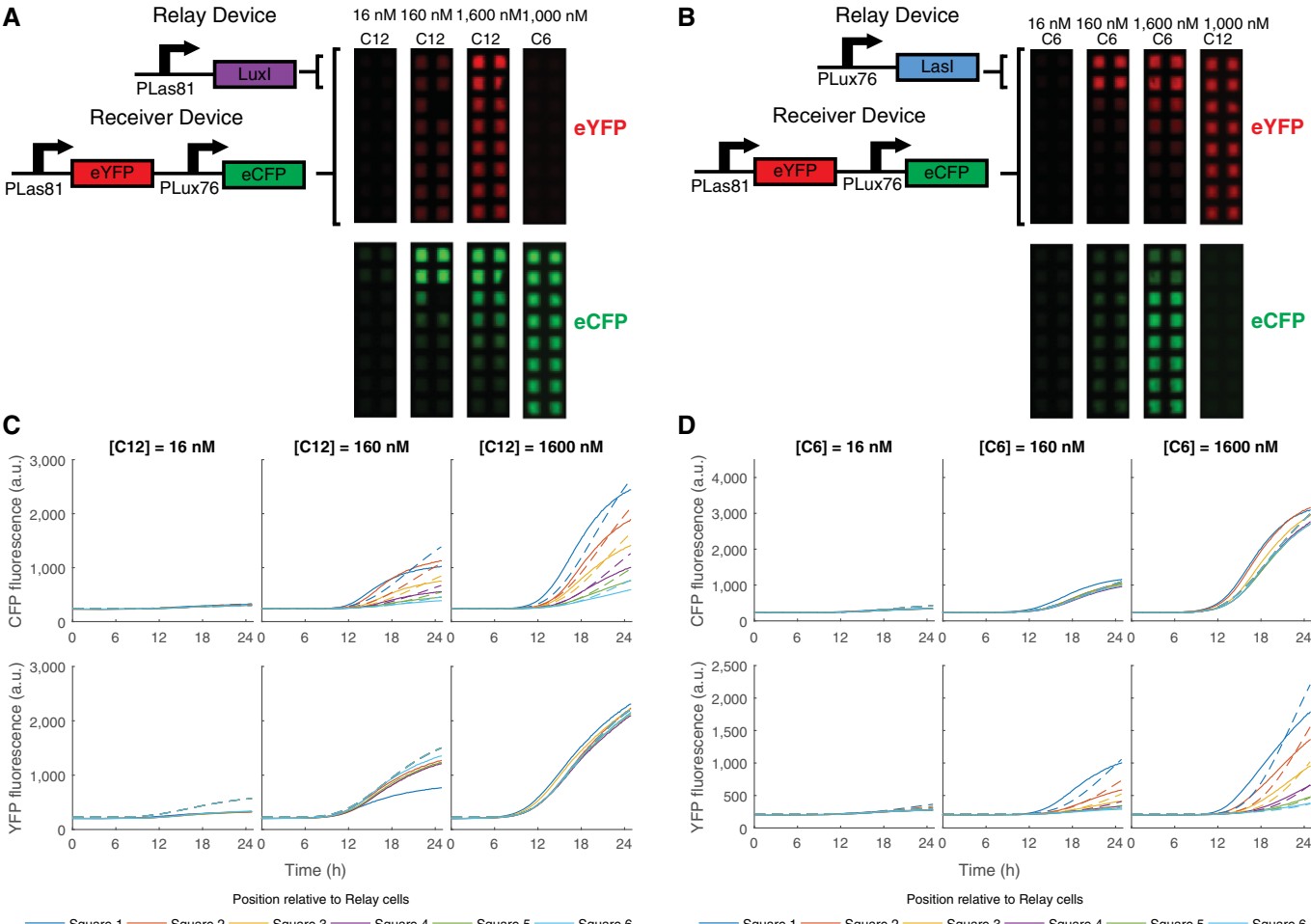

**Figure 4. Relay devices respond to one HSL signal by sending the other.**

A, B    Images at *t* = 1,500 min of chromosomal constitutive mRFP1 cells transformed with the pR33S175 double receiver and a relay device plated in 2 rows of 2 grid squares adjacent to cells containing the pR33S175 double receiver plus empty vector (for antibiotic resistance) in wells containing AHLs at the concentrations specified. (A)The pLas81LuxI relay device. (B) The pLux76LasI relay device.

C, D    eCFP (top) and eYFP (bottom) fluorescence (arbitrary units) plotted against time for the grid squares containing receiver plus empty vector. Solid lines plot data while dashed lines plot the model. Position relative to the grid squares containing relay devices are according to the color code shown. (C) The pLas81LuxI relay device. (D) The pLux76LasI relay device.

Source data are available online for this figure.

only in the YFP channel again showing no evidence of crosstalk in sending or receiving (Fig 4B).

### Autoinduction and long-range propagation of signals through alternating mutual activation

With all the components built, we used the cellular interactions we had engineered to create population-level behaviors. In the appropriate geometry, each relay device should be capable of activating the other, resulting in positive feedback through mutual activation. This behavior was used to create population arrangements capable of propagating a signal from one location to another. We arranged alternating stripes of cells containing each relay device along with the double receiver (Fig 5A). In this arrangement, a 3OC12HSL signal plated adjacent to the first stripe resulted in sending of 3OC6HSL by pLas81-LuxI relay cells. The 3OC6HSL signal then diffused to the adjacent pLux76-LasI stripe, resulting in sending of more 3OC12HSL, amplifying the original sending behavior and propagating the signal to the next stripe. In this way, the signal was propagated down the well by the successive activation of the sending behavior of each stripe (Movie EV1). When the pLas81-LuxI relay device was instead alternated with double receiver with empty vector (Fig 5B), the same signal resulted in a primary response in the YFP channel and a secondary response due to sending in the CFP channel, but the signal was not propagated. Similarly, the pLux76-LasI relay device alternated with double receiver with empty vector (Fig 5C) displayed only the primary response to signal in the YFP channel.

Changing the arrangement of these populations allowed signal initiation in addition to propagation. We plated relay devices plus double receivers alternating in a checkerboard arrangement (Fig 5E). This arrangement (which we will refer to as uninduced) was not sufficient to initiate positive feedback on its own (Fig 5E). However, when we additionally plated culture containing a 50:50 mixture of both cell types in a square in the center of the well (induced, Fig 5D), this mixed population initiated the mutual activation positive feedback loop. This was then propagated through the surrounding checkerboard pattern, resulting in strong expression in both the CFP and YFP channels (Movie EV2). Analysis of the system parameters revealed a bifurcation in behavior above certain values of signal production (either production of LuxI/LasI or synthesis rate of 3OC6HSL/3OC12HSL, Fig EV3B). At low signal production, the system remained "off", with minimal production of signal or fluorescent protein. Above the bifurcation point, the system eventually turned "on", allowing the positive feedback loop to occur, producing both signals and both fluorescent proteins. When parameters were sufficiently high for positive feedback, this feedback occurred first in the mixed population and then propagated through the

surrounding cells, as observed in the experiment (Fig EV3A). In this system, we have engineered behaviors at multiple levels of organization: we developed components that we have optimized at the molecular level using a combination of modeling, data collection, and rational design and used them to engineer intercellular interactions within and between populations. This results in population-level behaviors such as long-range signal propagation that are dependent on the geometry of those populations.

## Discussion

Engineering self-organizing multicellular systems requires control over interactions at multiple levels of organization. By using ratiometric measurements that allowed robust and reproducible quantification of the output of genetic devices, we were able to characterize the molecular interactions that led to crosstalk between the 3OC6HSL and 3OC12HSL signaling systems. We realized that changes in the expression level of the receiver proteins LuxR and LasR had a strong effect on the level of chemical crosstalk. We were able to turn this insight into quantitative understanding by building a mathematical model of the entire system. We could then make changes to the expression levels of receiver proteins and use that data to successfully infer the large number of parameters that were required for a mechanistic model of the system.

We addressed genetic crosstalk by screening a small number of rationally designed variants of the pLux promoter. By making changes in base pairs predicted to be required for LasR binding but not LuxR binding, we were able to create a Lux-specific promoter that greatly reduced the affinity of LasR, while maintaining affinity for LuxR. Conversely, we were able to turn pLux into a Las-specific promoter by making base pair changes that not only reduced the affinity for LuxR but also increased the affinity for LasR. We could then use these two promoters in a two-channel receiver device to create orthogonal responses to 3OC6HSL and 3OC12HSL, in the context of the appropriate expression level of the receiver proteins. These promoters will serve as useful new parts in the creation of novel synthetic signaling circuits. We used our model to predict the optimal expression level that would maximize the ratio of signal to crosstalk. We also used the model to infer the expression level of each receiver protein in newly created devices, by measuring their responses to 3OC6HSL and 3OC12HSL in each of the fluorescent channels. This gave us a target, together with a method for evaluating how close we came to hitting that target, allowing us to arrive at an optimal device with only a small number of iterations.

We see this as a generalizable method for optimizing the expression of circuit components. By measuring devices, making small

---

**Figure 5. Relay devices can initiate and propagate signals through population-level positive feedback.**

A–E   All cells express chromosomal constitutive mRFP1 and are plated on membranes printed with hydrophobic grids. Cells transformed with constitutive control (pPRYFPPRCFP) and plated along the edge of the membrane provide a standard for normalization and minimize edge effects. (A–C) Images at $t = 1,500$ min of relay devices co-transformed with double receiver (pR33S175) and arranged in alternating stripes with a width of two grid squares. Blue arrows indicate stripes of pLux76LasI while purple arrows indicate pLas81LuxI. Stripes with no arrow contain double receiver with empty vector. About 15 µl of 20 µM 3OC12HSL was plated in a stripe above the first stripe of cells at $t = 0$. (A) Alternating stripes of pLux76LasI and pLas81LuxI. (B) Alternating stripes of pLas81LuxI and empty vector. (C) Alternating stripes of pLux76LasI and empty vector. (D) Images at the time points indicated of cells co-transformed with double receiver (pR33S175) and either pLux76LasI or pLas81LuxI and plated in a checkerboard pattern with the addition of a mixed population of both cell types in an 8 × 8 square in the center. (E) Images at the timepoints indicated of a layout identical to (D) except for the absence of a mixed population.

Source data are available online for this figure.

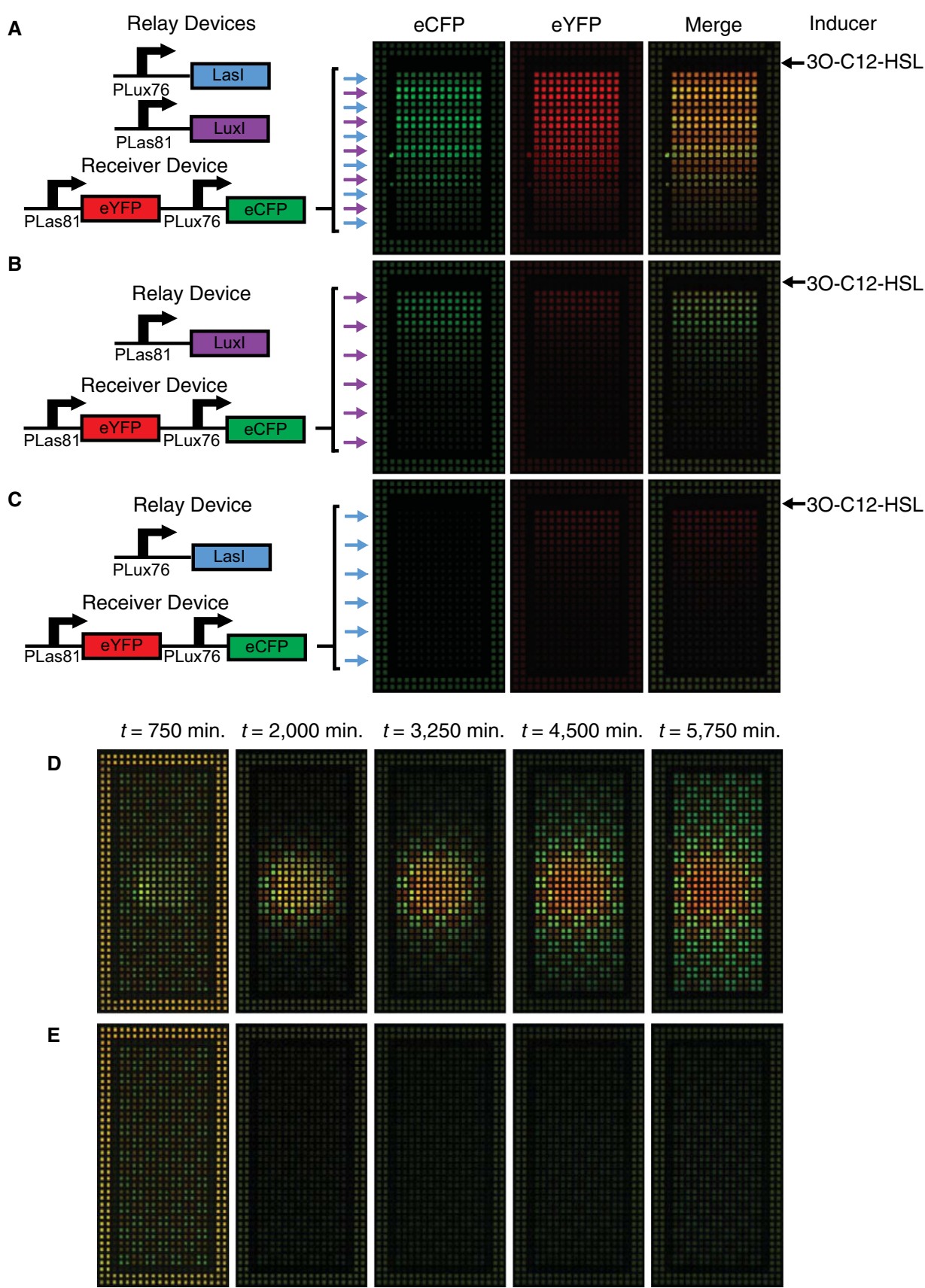

**Figure 5.**

changes, and measuring again, one can build up a complete picture of a system. This can be formalized in a quantitative model that provides both a check on the internal consistency of the data, together with accurate predictions about how future designs will behave. The model also allows inference about circuit components that are not directly measured, which means the expression level of those components can be tuned in the context of the circuit in which they are functioning. This method allows predictable engineering even with parts that are imperfectly modular. This is important as attempting to diagnose and control the effects of genetic context is still work in progress and can require screening many variants (Lou *et al*, 2012; Kosuri *et al*, 2013; Mutalik *et al*, 2013a,b). Another possible approach would be to use directed evolution techniques such as MAGE (Wang *et al*, 2009) to select for variants that achieve better orthogonality but our approach provides the added benefit of creating useful knowledge about the underlying behaviors of a system. By building up a model of a circuit functioning in the context for which it is designed, we are able to continue to build on existing devices and provide a basis for the design of more complex circuits.

The test of the extent of our understanding was to move our devices from liquid culture to a novel solid culture system and be able to model their behavior by inferring only the new parameters of the system such as signal production and diffusion. This system, consisting of membranes printed with hydrophobic grids placed on solid media and imaged in an incubated macroscopic imaging chamber, provides a simple way of arranging and maintaining spatially discrete populations in arbitrary geometries. By using a chromosomally integrated mRFP1 signal as a proxy for growth, fluorescent output from these populations can be measured ratiometrically in both a plate fluorometer and on membranes. This allows us to quantitatively analyze the spatial behavior of circuits and translate knowledge gained from liquid culture experiments to spatially organized populations on solid media. This control and reproducibility over space allowed us to engineer precise population-level interactions.

Engineering population-level interactions opens new avenues of investigation. We have successfully engineered an intercellular positive feedback loop whose behaviors can be tuned not only by modifying its genetic components but also by changing the geometric arrangement of the populations involved. By changing from a mixed population to a checkerboard arrangement, the system switches from autoinduction to signal propagation. This type of system in which genetics and spatially organized population interactions contribute to the overall system behavior opens up new approaches to designing systems that possess interesting dynamics, perform information processing, and self-organize. Using the parts, devices, and techniques we have developed here, we are poised to create systems that use the spatiotemporal patterning properties of intercellular signaling to organize matter at scales from the molecular to the macroscopic.

# Materials and Methods

### Plasmid construction

All plasmids (listed in Appendix Tables S1–S3) were constructed using Gibson assembly (Gibson *et al*, 2009) from parts obtained from the MIT Registry of Standard Biological Parts (http://partsregistry.org) or synthesized by DNA 2.0 (Menlo Park, CA, USA) and

are available on Addgene (www.addgene.org). Sequences are available on Genbank (accession numbers KU523969, KU523970, KU523971, KU523972 and KU523973). Ratiometric receiver devices were based on devices described previously (Yordanov *et al*, 2014). Relay devices were cloned into the pSB6A1 backbone (http://partsregistry.org). All cloning and analysis was performed in *E. coli* strain *E. cloni 10G* (Lucigen).

### Plate fluorometer assays

Plate fluorometer assays and data analysis were conducted as previously described (Yordanov *et al*, 2014). Overnight cultures were diluted 1:1,000 in M9 medium supplemented with 0.2% casamino acids and 0.4% glucose in a volume of 200 µl per well and measurements taken every 10 min for 1,000 min in a BMG FLUOstar Omega plate fluorometer. 3-oxohexanoyl-homoserine lactone (3OC6HSL, Cayman Chemicals) and 3-oxododecanoyl-homoserine lactone (3OC12HSL, Sigma) were dissolved to a concentration of 200 mM in DMSO then 3OC6HSL was diluted in M9 medium supplemented with 0.2% casamino acids and 0.4% glucose to the concentrations described, while 3OC12HSL, due to its limited solubility in aqueous media, was first diluted 1:50 in ethanol then diluted in supplemented M9 medium to the concentrations described.

### Solid culture assays

Single colonies were picked from LB agar plates and grown overnight in supplemented M9 medium with appropriate antibiotics (50 µg/ml kanamycin, 50 µg/ml carbenicillin). Cultures were diluted 1:100 then grown into exponential phase (2–4 h) and rediluted to an optical density at 600 nM of 0.05. This dilute culture was spotted onto Iso-Grid membranes (Neogen, Lansing, MI, USA) placed on 1.5% agar plates containing the same supplemented M9 growth medium. The culture was plated at a volume of 0.5 µl per grid square. Plates were imaged in a custom imaging device consisting of an optical breadboard and frame (Thorlabs) on which were mounted LED lightsources (Amber [591 nm], Cyan [505 nm], Royal-Blue [447.5 nm], and Luxeon Rebel Star CoolBase LEDs). The output was collimated with a lens (Carlco), filtered with excitation short pass filters (Comar Instruments) of 581 nm, 510 nm, and 450 nm, respectively, and shaped by engineered top hat diffusers (Thorlabs, ED1-C50-MD). A monochromatic camera (Photometrics, CoolSNAP HQ2) with a zoom lens and 10-nm bandpass filters (Edmund Optics) of 636 nm, 540 nm, and 486 nm, respectively, in a filter wheel was used to collect the emission. Walls were constructed of light-tight cardboard and temperature was maintained at 37°C using an air-powered microscope stage incubator (Nevtek). To maintain humidity, samples were placed within a chamber containing a water reservoir and fitted with a glass lid. LEDs were powered using an Arduino duemilanove microcontroller (Arduino) and controlled in concert with the filter wheel using Micromanager.

### Chromosomal constitutive mRFP1

mRFP1 (BBa_E1010) driven by the lambda phage PR promoter (BBa_R0051) was cloned into the "landing pad" region of pTKS/CS that included a tetracycline resistance cassette (Kuhlman & Cox, 2010). Linear DNA containing the landing pad region was amplified using primers to add 50 bp of homology to the coding region of the

arsB gene (Sabri *et al*, 2013) on each end of the landing pad. The landing pad was incorporated into the genome at the arsB locus using Red/ET recombination (GeneBridges), selected by tetracycline resistance and confirmed by sequencing.

Expanded View for this article is available online.

## Acknowledgements

PKG acknowledges support from the John Templeton Foundation Grant ID#15619: "Mind, Mechanism and Mathematics: Turing Centenary Research Project". JH acknowledges Biotechnology and Biological Sciences Research Council and Engineering and Physical Sciences Research Council (RG72490), and FF acknowledges support from CONICYT-PAI/Concurso Nacional de Apoyo al Retorno de Investigadores/ as desde el Extranjero Folio 82130027. We would like to thank J. Ajioka and O. Yarkoni for use of equipment and advice. We would like to thank P.J. Steiner for early discussions about this work.

## Author contributions

PKG, ND, JRB, AP, and JH designed the research. PKG, FF, OP, and TJR performed the experiments. PKG, ND, TJR, and BY analyzed the data. ND and BY performed computational modeling. PKG, ND, AP, and JH wrote the manuscript.

## Conflict of interest

The authors declare that they have no conflict of interest.

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
