## [Review Process File · Molecular Systems Biology]

Orthogonal intercellular signaling for programmed spatial behaviour

Paul K. Grant, Neil Dalchau, James R. Brown, Fernan Federici, Timothy J. Rudge, Boyan Yordanov, Om Patange, Andrew Phillips and Jim Haseloff

Corresponding author: Jim Haseloff, University of Cambridge

Review timeline:

Submission date:	18 September 2015
Editorial Decision:	22 October 2015
Revision received:	19 November 2015
Accepted:	23 November 2015

Editor: Thomas Lemberger

Transaction Report:

1st Editorial Decision

22 October 2015

Thank you again for submitting your work to Molecular Systems Biology. We have now heard back from the three referees who accepted to evaluate the study. As you will see, the referees find the topic of your study of potential interest and they are clearly supportive. They make nevertheless some suggestions for modifications and clarifications, which we would ask you to carefully address in a revision of the present work.

REFeree COMMENTS

Reviewer #1:

In this manuscript, the authors describe a method for tuning quorum sensing systems for use in synthetic biology. They are particularly interested in creating two orthogonal signaling pathways that can be used simultaneously with minimum crosstalk. The authors achieve this through promoter tuning guided by a mathematical model. Further, they demonstrate their approach by building "relay devices" that sense one signal and in response send out another.

Overall, the quality of the work is high and the presentation is good. Though I found it hard to find many major flaws with the work, there are some things that I think should be addressed.

1) Unfortunately, the novelty of the work is tempered by the recent publication of a synthetic system that used two orthogonal quorum sensing systems to create oscillations in a microbial consortium (Chen et al, Science 349, 986 (2015)). In that work, the authors used the rhlI/R system (C4HSL) and the cinI/R system (3-OHC14HSL), which work orthogonally in E coli. Therefore, the statement

made by Grant et al in the present manuscript that "we have, for the first time, created a device that differentiates between two different AHL inputs in the same cell..." should probably be reworded. Of course, the ability to engineer the C6 and C12 systems to be orthogonal is still a big deal and very helpful. But, it should be put in context with other recent advances.

2) It would be a big help if, in addition to the promoter gene cartoons in figures, the authors could provide a regulatory map of the interactions in the cells, especially in the last three figures.

3) The spatial assay method is interesting. A figure in the main text that helps to explain the method for the uninitiated would be a big help. Similarly, it's hard to tell exactly what is going on in Fig. 5 -- i.e. its not quite clear what the blue and purple arrows represent at first glance. In 5D and E, a visual explanation of the spatial arrangement of the cell types would also help.

4) As written in the main text, the mathematical modeling is somewhat hard to follow. Its almost as if the authors could simply say "the model predicted we should do X to get better orthogonality (See SI), and so we did that..." Alternatively, a clearer and more detailed description of the model would help - but I am not sure that is necessary.

Reviewer #2:

This paper by Grant and colleagues demonstrates one of the first approaches toward engineering orthogonality into quorum sensing systems from bacteria. Early studies on these quorum sensing systems, such as those by Gray et al. 1994 (*J. Bacteriology*), pointed out the inherent cross-talk between these systems, limiting the possible designs of future gene circuits using multiple QS components (especially within a single cell). This work takes a systematic approach toward the problem by investigating each of these cross-talk interactions individually and taking rational steps toward mitigating their effects, namely, by tuning protein expression and promoter binding affinity. The use of a computational model, informed by experiments, highlights the rational design aspect of the paper and may be applicable to future studies on cross-talk interactions with different QS systems.

Putting both systems in a single cell, the authors are able to apply spatial gene expression control using the respective AHL signaling molecules (Figure 3), highlighting the orthogonality of the system and its applicability to engineering spatial gene expression patterns. I think, however, that the authors may want to provide some references to highlight the importance of bidirectional signaling in biological systems to better highlight the significance of the patterning experiments (using the relay set-up) in Figures 4 and 5.

Overall this is a great paper with well detailed text and figures to guide the reader through the design process and represents a significant advance for orthogonal QS-based gene circuits. This work is an excellent example of the engineering approach to synthetic biology often described but rarely implemented. Furthermore, the authors address a problem that has been much-lamented but never addressed in synthetic biology: the lack of multiple orthogonal quorum sensing systems.

On the whole this work is interesting and described in (much appreciated) detail. Here are a few minor points to address:

I'm not sure how the 'mutually activating' signal relay circuit accomplishes 'signal relay' in a way that is more meaningful than simple positive feedback in a uniform population. Is there some advantage to this more complicated configuration beyond as a demonstration of more complicated circuits and the ability to arrange populations in space? Perhaps there is something different about the dynamics of signal propagation?

I recognize that you mention C6 is 3-O-C6, and C12 is 3-O-C12, but given the existence of actual C6 in addition to 3OC6 signals I think this might be confusing.

The authors should provide the culturing and measurement times for the plate fluorometer assays in the Materials and Methods, as well as the instruments used for expression measurement.

Figure 5, panel E is missing a description on the figure caption.

Results, first paragraph

"requiring us to use a new reference pLlacO1 (Bba_R0051)"

Is this a typo? The number and an earlier sentence in the paragraph suggest that the reference is PR.

'A mathematical model of signal crosstalk', first paragraph

"For this Full model (SI Section B), it was not possible to derive a closed-form expression (a single equation purely as a function of the parameters) for the transcription rate."

I don't think it's necessary to italicize 'closed-form' here.

Reviewer #3:

Thank you for inviting me to review this manuscript. My own expertise is in molecular biology and experimental implementation of synthetic biology. The bulk of the work described in this manuscript is therefore relevant to my expertise, although I am not able to provide critique of the mathematical modelling aspects of this manuscript.

The manuscript describes an impressive and thorough set of experiments and accompanying models that lead to *E. coli* bacteria that are rationally programmed to produce and detect two different chemical signals (both AHLs) with minimum crosstalk between the two. This ability is crucial for enabling pattern formation, and is a key step towards generating Turing patterns, which require two different diffusible chemical signals. With the bacteria engineered to produce and detect the two different chemical signalling molecules, the manuscript shows how this can be used to generate and propagate patterns in novel 'printed' bacterial colony systems. The videos attached to the manuscript are very impressive to watch and nicely illustrate the achievement of this work, both in terms of bacterial engineering by synthetic biology and also the novel method of using printed colonies and high quality fluorescence microscopy. However, the real tour-de-force of this work is the intertwined experimental and modelling work that was used to optimise the production and detection of the two AHL chemical signals so that crosstalk was minimized. Previous works have attempted pattern formation with AHL signalling but have always been limited by the acute crosstalk between different AHL molecules and their intracellular receptors/promoters. This manuscript sets a new high bar for rational, model-led optimisation in synthetic biology and does so precisely to tackle this crosstalk issue. Altogether, this is an impressive body of interdisciplinary work that showcases what is possible in synthetic biology.

I highly recommend publication of this manuscript, which can be seen as a significant advance in several areas. In synthetic biology in general, the success of the iterative exchange in this manuscript between modelling and experimental work is very thorough and particularly impressive. It is beyond anything I've seen before. For engineering pattern formation by programming bacteria, the sender and relay system demonstrated at the end of this manuscript are an advance on the key papers in synthetic biology pattern formation. Also the methodology of visualising pattern propagation is novel and will be of great value to the field. The work will obviously be of great interest to those interested in engineering pattern formation either by experimental methods or by mathematical design and simulation. Beyond this however, I feel this will also be a key paper for synthetic biology and of great interest also to those in mathematical biology and systems biology too. It appeals directly to all of those in these fields that are interested in combining mathematical approaches with biological experimentation and may well become the exemplar paper in this regard.

While I myself cannot critique the modelling aspects of this manuscript (and its considerable supplementary material) with an expert eye, the overall work and particularly the experimental implementation appear largely sound to me and I don't think I can see any reasons for major revisions. As expected of an MSB manuscript, the work is already described in considerable length and depth. I do have some minor comments and suggestions however.

1. Microscopy figures - all fluorescence imaging figures use false colouring and have chosen red for yellow fluorescent protein, green for cyan fluorescent protein and blue for red fluorescent protein. This is quite counterintuitive. As a reader, flicking between descriptions of RFP expression and photos of blue colonies is a confusing task. Why not use the colours of the corresponding fluorescent proteins?
2. Figures 1C and 2C and 2D - why does AHL-C12 not get measured at the same high concentrations as AHL-C6? I may have missed this but it makes the graphs look suspicious to have one dataset terminate early.
3. Colony printing images in Figures - without a bright field image of the colony (start or finish) it is hard to get an understanding of what the layout of colonies is supposed to be, or how the colony printing works. Please add bright-field images.
4. Discussion - The authors discuss how their work gives a guide to others in synthetic biology and so helps the field. They should add to this that their designed promoters will also be valuable as high-quality basic parts for further *E. coli* synthetic biology projects.
5. Discussion - One of the most impressive aspects of this manuscript is the use of modelling to optimise the reduction of crosstalk in the system. However, this could arguably have been achieved by instead using clever directed evolution. Can the authors add to the discussion an evaluation of how their approach compares to instead disregarding modelling and utilising large-scale mutation/screening approaches to select Lux/Las and promoter variants with minimised crosstalk?
6. Supplementary Materials - references to the supplementary materials in the main manuscript are often confusing or incorrect. I think on a couple of occasions I was directed to the wrong Supplementary Figure. Please can the authors check this.
7. Supplementary Figure 9 - this Figure intrigues me! It is barely mentioned in the main manuscript and has a very brief figure legend in the supplementary, so I can't really determine what each graph is supposed to represent. However, the data points in two of the five graphs in part B appear to show 'pulse/peak' behaviour although the lines seem to ignore this. What are these and why could they be showing peaks of expression? Is this relevant?

1st Revision - authors' response

19 November 2015

Response to reviewer's comments:

Reviewer #1:

1) Unfortunately, the novelty of the work is tempered by the recent publication of a synthetic system that used two orthogonal quorum sensing systems to create oscillations in a microbial consortium (Chen et al, Science 349, 986 (2015)). In that work, the authors used the rhII/R system (C4HSL) and the cinI/R system (3-OHC14HSL), which work orthogonally in E coli. Therefore, the statement made by Grant et al in the present manuscript that "we have, for the first time, created a device that differentiates between two different AHL inputs in the same cell..." should probably be reworded. Of course, the ability to engineer the C6 and C12 systems to be orthogonal is still a big deal and very helpful. But, it should be put in context with other recent advances.

We agree that the work of Chen et al. is significant and have included a reference to this paper (page 3, paragraph 2). Crosstalk is not directly measured in their paper so we are unsure to what extent these signals are truly orthogonal. Nonetheless, Chen et al. have produced a practical working system. The reworded sentence now reads (page 3, paragraph

2): “In doing so, we have systematically reduced crosstalk to produce a device that differentiates between two different AHL inputs in the same cell and produces two orthogonal outputs.”

2) It would be a big help if, in addition to the promoter gene cartoons in figures, the authors could provide a regulatory map of the interactions in the cells, especially in the last three figures.

We have added figure EV3 that contains the regulatory maps requested (A-C).

3) The spatial assay method is interesting. A figure in the main text that helps to explain the method for the uninitiated would be a big help. Similarly, it's hard to tell exactly what is going on in Fig. 5 -- i.e. its not quite clear what the blue and purple arrows represent at first glance. In 5D and E, a visual explanation of the spatial arrangement of the cell types would also help.

Figure EV3 contains a photograph of bacterial populations growing on membranes (D) as well as diagrams showing the layout of the cells in figures 4 and 5 in order to make the experimental setup more clear.

4) As written in the main text, the mathematical modeling is somewhat hard to follow. Its almost as if the authors could simply say "the model predicted we should do X to get better orthogonality (See SI), and so we did that..." Alternatively, a clearer and more detailed description of the model would help - but I am not sure that is necessary.

We have provided a more detailed description of the model in the main text (page 4, paragraph 4 through page 5, paragraph 3).

Reviewer #2:

Putting both systems in a single cell, the authors are able to apply spatial gene expression control using the respective AHL signaling molecules (Figure 3), highlighting the orthogonality of the system and it's applicability to engineering spatial gene expression patterns. I think, however, that the authors may want to provide some references to highlight the importance of bidirectional signaling in biological systems to better highlight the significance of the patterning experiments (using the relay set-up) in Figures 4 and 5.

The following sentence has been added to the introduction (page 2, paragraph 3):

“Multicellular patterning mechanisms such as those proposed by Turing (Turing, 1952) and Gierer and Meinhardt (Gierer & Meinhardt, 1972) as well as the creation of tissue organizing centers (Spemann & Mangold, 1924; Struhl & Basler, 1993) all require bidirectional signaling between populations of cells.”

I'm not sure how the 'mutually activating' signal relay circuit accomplishes 'signal relay' in a way that is more meaningful than simple positive feedback in a uniform population. Is there some advantage to this more complicated configuration beyond as a demonstration of more complicated circuits and the ability to arrange populations in space? Perhaps there is something different about the dynamics of signal propagation?

We believe that mutual activation of distinct populations opens up new mechanisms for tuning the system based on the geometry of the populations and may provide a more robust mechanism for creating signal propagation

in space. A simple positive feedback in a uniform population would require extremely precise tuning of genetic parameters such that the system remains off in the absence of signal even as a population grows in cell number yet maintains the ability to activate and propagate an applied signal. By separating the components of the feedback loop into two populations it is easier to keep the loop “off” in the absence of signal and activation can be tuned by the spacing of the populations (as in Figure 5D-E).

I recognize that you mention C6 is 3-O-C6, and C12 is 3-O-C12, but given the existence of actual C6 in addition to 3OC6 signals I think this might be confusing.

We have changed the abbreviations to 3OC6HSL and 3OC12HSL throughout.

The authors should provide the culturing and measurement times for the plate fluorometer assays in the Materials and Methods, as well as the instruments used for expression measurement.

We have added the following sentence to Materials and Methods (page 14, paragraph 3):

“Overnight cultures were diluted 1:1000 in M9 medium supplemented with 0.2% casamino acids and 0.4% glucose in a volume of 200µL per well and measurements taken every 10 minutes for 1000 minutes in a BMG FLUOstar Omega plate fluorometer.”

Figure 5, panel E is missing a description on the figure caption.

We thank the reviewer for pointing this out and have corrected this (page 21).

Results, first paragraph

"requiring us to use a new reference pLlacO1 (Bba_R0051)"

Is this a typo? The number and an earlier sentence in the paragraph suggest that the reference is PR.

The reviewer is correct and we have corrected the typo to reflect that the promoter used was pR (page 4, paragraph 1).

'A mathematical model of signal crosstalk', first paragraph

"For this Full model (SI Section B), it was not possible to derive a closedform expression (a single equation purely as a function of the parameters) for the transcription rate."

I don't think it's necessary to italicize 'closed-form' here.

We have removed the italics (page 5, paragraph 2).

Reviewer #3:

1. Microscopy figures - all fluorescence imaging figures use false colouring and have chosen red for yellow fluorescent protein, green for cyan fluorescent protein and blue for red fluorescent protein. This is quite counterintuitive. As a reader, flicking between descriptions of RFP expression and photos of blue colonies is a confusing task. Why not use the colours of the corresponding fluorescent proteins?

We feel that using red and green for the important channels (YFP and CFP) allows the best visualization of both signal strength and degree of contribution of each channel. Here for comparison is a timepoint from figure 5 with cyan and yellow channels which we feel is more difficult to interpret

than the red and green images:

2. Figures 1C and 2C and 2D - why does AHL-C12 not get measured at the same high concentrations as AHL-C6? I may have missed this but it makes the graphs look suspicious to have one dataset terminate early.

3OC12HSL is less soluble than 3OC6HSL in aqueous media so we were unable to make solutions of higher concentration using our dilution procedures but wanted to capture the response to 3OC6HSL at those high concentrations. We have included this detail in Materials and Methods (page 14, paragraph 3).

3. Colony printing images in Figures - without a bright field image of the colony (start or finish) it is hard to get an understanding of what the layout of colonies is supposed to be, or how the colony printing works. Please add bright-field images.

We have added Figure EV3 which contains a photograph of cell populations growing on filters. Such a photograph will look the same for all experiments as what differs is only the FP expression but we have included diagrams in Figure EV3 that explain the layout of the cell populations in experiments depicted in Figures 4 and 5.

4. Discussion - The authors discuss how their work gives a guide to others in synthetic biology and so helps the field. They should add to this that

their designed promoters will also be valuable as high-quality basic parts for further E. coli synthetic biology projects.

We have added the following sentence to the discussion (page 13, paragraph 1):

“These promoters will serve as useful new parts in the creation of novel synthetic signaling circuits.”

5. Discussion - One of the most impressive aspects of this manuscript is the use of modelling to optimise the reduction of crosstalk in the system. However, this could arguably have been achieved by instead using clever directed evolution. Can the authors add to the discussion an evaluation of how their approach compares to instead disregarding modelling and utilising large-scale mutation/screening approaches to select Lux/Las and promoter variants with minimised crosstalk?

We have added the following sentences to the discussion (page 13, paragraph 2):

Another possible approach would be to use directed evolution techniques such as MAGE (Wang *et al*, 2009) to select for variants that achieve better orthogonality but our approach provides the added benefit of creating useful knowledge about the underlying behaviors of a system. By building up a model of a circuit functioning in the context for which it is designed, we are able to continue to build on existing devices and provide a basis for the design of more complex circuits.

6. Supplementary Materials - references to the supplementary materials in the main manuscript are often confusing or incorrect. I think on a couple of occasions I was directed to the wrong Supplementary Figure. Please can the authors check this.

We have corrected the numbering to reflect the addition of Expanded View figures and ensured the references are correct in the main text.

7. Supplementary Figure 9 - this Figure intrigues me! It is barely mentioned in the main manuscript and has a very brief figure legend in the supplementary, so I can't really determine what each graph is supposed to represent. However, the data points in two of the five graphs in part B appear to show 'pulse/peak' behaviour although the lines seem to ignore this. What are these and why could they be showing peaks of expression? Is this relevant?

We have added a reference to this figure in the main text (page 9, paragraph 2) and added the following text to the supplement:

“Constructs in which LasR translation was regulated by RBS Bba_B0034 (pR100S34, pR33S34) displayed a transfer function whose shape was not well fit by the model. The presence of a peak at lower concentrations of C12, dipping to a lower plateau at higher concentrations suggests a regulatory mechanism which our model does not account for, and that is revealed only when LasR is expressed at very high levels.”

We agree with the reviewer that the shapes of the transfer functions for constructs containing LasR under RBS Bba_B0034 are anomalous. We only observe this “plateau” shape in conditions of very high expression. This remains to be investigated.